# Chemical control of excited-state reactivity of the anionic green fluorescent protein chromophore

Nanna H. List [1✉], Chey M. Jones[2,3] & Todd J. Martínez [2,3✉]

Controlling excited-state reactivity is a long-standing challenge in photochemistry, as a desired pathway may be inaccessible or compete with other unwanted channels. An important example is internal conversion of the anionic green fluorescent protein (GFP) chromophore where non-selective progress along two competing torsional modes (P: phenolate and I: imidazolinone) impairs and enables Z-to-E photoisomerization, respectively. Developing strategies to promote photoisomerization could drive new areas of applications of GFP-like proteins. Motivated by the charge-transfer dichotomy of the torsional modes, we explore chemical substitution on the P-ring of the chromophore as a way to control excited-state pathways and improve photoisomerization. As demonstrated by methoxylation, selective P-twisting appears difficult to achieve because the electron-donating potential effects of the substituents are counteracted by inertial effects that directly retard the motion. Conversely, these effects act in concert to promote I-twisting when introducing electron-withdrawing groups. Specifically, 2,3,5-trifluorination leads to both pathway selectivity and a more direct approach to the I-twisted intersection which, in turn, doubles the photo-isomerization quantum yield. Our results suggest P-ring engineering as an effective approach to boost photoisomerization of the anionic GFP chromophore.

[1] Department of Chemistry, School of Engineering Sciences in Chemistry, Biotechnology and Health, KTH Royal Institute of Technology, SE-10044 Stockholm, Sweden. [2] Department of Chemistry and The PULSE Institute, Stanford University, Stanford, CA 94305, USA. [3] SLAC National Accelerator Laboratory, 2575 Sand Hill Road, Menlo Park, CA 94025, USA. ✉email: nalist@kth.se; toddjmartinez@gmail.com

The green fluorescent protein (GFP[1–4]) and its relatives have tremendous impact on life sciences by making the invisible visible[5–8]. At their structural and functional core lies the monomethine chromophore motif, 4-hydroxybenzylidene-2,3-dimethylimidazolinone (HBDI), which is responsible for light absorption and emission[9]. Its high tunability in response to mutations of the protein β-barrel or chemical modifications enables a remarkable range of photoinduced behaviors (e.g., fluorescence, photooxidation, and photoisomerization)[10–13]. Understanding the link between these molecular control variables and the fate of the photoinduced nonequilibrium state could enable systematic protein design that factors in photoinduced function. The present work focuses on photoisomerization.

The anionic HBDI chromophore (HBDI⁻, Fig. 1) undergoes ultrafast internal conversion mediated by competing torsional motion around either the imidazoline (I) or the phenolate (P) bond of the central methine bridge (indicated by curly arrows in Fig. 1)[14]. I-torsional motion can lead to productive isomerization (i.e., generating E-stereoisomer), whereas rotation around the (phenolate) P-bond regenerates the original ground-state Z-stereoisomer. While the extent of photoisomerization for the isolated chromophore remains to be experimentally resolved, our recent nonadiabatic dynamics simulations indicate a non-negligible Z-to-E quantum yield of ~30%[15]. Photoisomerization is partially retained inside the protein scaffolds of reversibly photoswitchable variants, although the on-to-off (corresponding to Z-to-E) photoisomerization quantum yields are generally very low (<1%)[16]. A lowered trend is also found in aqueous solution with a Z-to-E quantum yield <5%[17] (~10% in our previous simulations[18]). Nevertheless, the photoisomerization within a protein setting allows for optical switching of the fluorescence, enabling super-resolution imaging[12]. Improving their photo-isomerization is not only relevant for developing next-generation passive reporters for nanoscopy but could also offer exciting possibilities in active-control applications. For example, Boxer and coworkers discovered that light can activate strand dissociation of a circularly permuted split GFP into peptide strand and truncated protein, which could be exploited as a non-neural optogenetic tool. However, one of the key factors limiting overall strand photodissociation is the low quantum yield of the gateway photoisomerization process[19–21].

We have previously identified intrinsic bottlenecks of the chromophore that need to be overcome to enhance photo-isomerization under isolated conditions (Fig. 2a)[15]. We found an almost equal branching ratio between the competing I- and

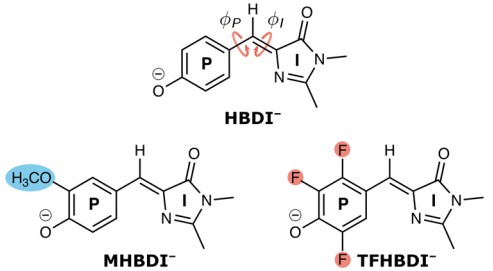

**Fig. 1 Chemical structures of the anionic GFP chromophores studied.** The unmodified chromophore (HBDI⁻) together with its 3-methoxylated (MHBDI⁻) and 2,3,5-trifluorinated (TFHBDI⁻) derivatives. The pertinent bridge P- and I-bonds (P: phenolate and I: imidazolinone) and associated torsions, $\phi_P$ and $\phi_I$, are indicated by curly arrows. The arrow directions define the positive rotation for the bridge dihedrals with zero angles corresponding to the Z-isomer. Definitions of key geometric parameters are provided in Supplementary Fig. 1.

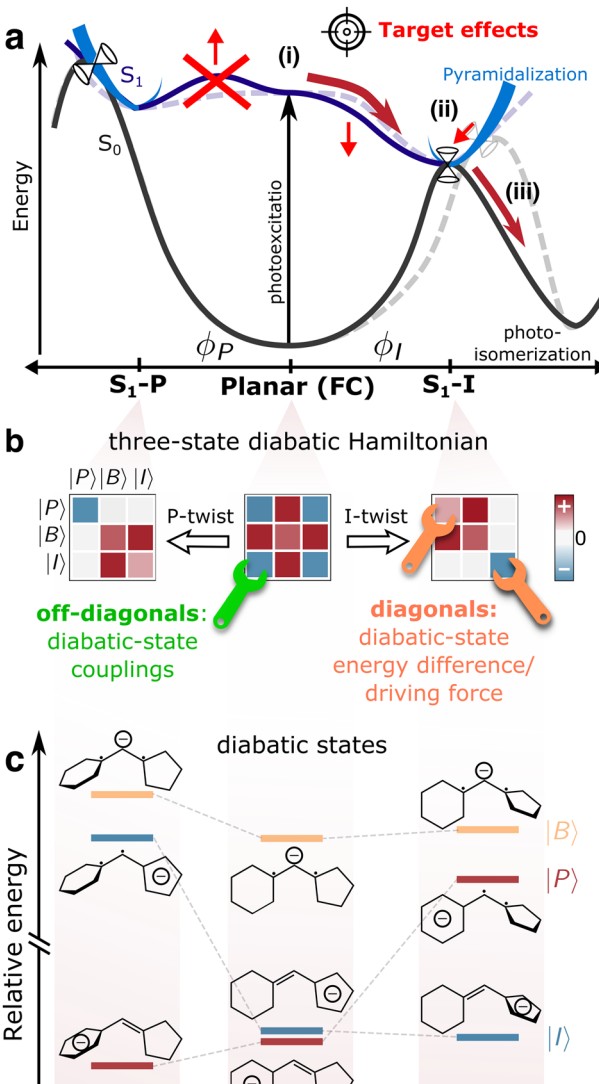

**Fig. 2 Required changes to promote photoisomerization in HBDI⁻ and available tuning knobs. a** Schematic of the targeted changes of the internal conversion dynamics. Dashed lines represent the ground- and excited-state potential energy curves along the bridge torsions (P: phenolate and I: imidazolinone) in HBDI⁻ (pyramidalization shown as blue curved lines) while solid lines represent the desired modification with the targeted changes indicated by the red arrows. In essence, the targeted effects are: (i) a bond-selective departure from the Franck–Condon (FC) point along the reactive I-twist channel; (ii) a more direct approach to the I-twisted intersection seam to increase the internal conversion efficiency and, in turn, (iii) increase the photoisomerization quantum yield (thick, dark red arrows). Reaching the conical intersection seams (cones) in HBDI⁻ requires pyramidalization of the methine bridge, which is energetically uphill relative to the twisted minima. **b** Schematic of the three-state diabatic Hamiltonian, which upon diagonalization gives the adiabatic states in (**a**). Progress along the bridge torsions leads to block-diagonal structures. The colored shading indicates the relative sign and magnitude of the matrix elements (diagonal: diabatic-state energies, off-diagonal: diabatic-state couplings). In the diabatic picture, substituents can affect the driving force (energy difference between diabatic states) and diabatic state couplings. **c** Diabatic state energies (i.e., the diagonal elements of the effective Hamiltonians in (**b**)), their charge distribution and bonding character across the bridge.

P-torsional pathways upon departure from the Franck–Condon (FC) point but with most of the population transfer (75%) occurring along the sloped I-twisted conical intersection seam. Access to the intersection seams is further gated by pyramidalization of the methine bridge, which dynamically is mediated mainly by methine hydrogen-out-of-plane (HOOP) motion (blue curved perspective lines, Fig. 2a). We further showed the ~30% photoproduct generation to be facilitated by a combination of early ballistic motion at a less reactive part of the I-twisted intersection seam (denoted MECI-I2$^+$) and near-statistical behavior around a more reactive region (denoted MECI-I$^+$)[15]. These findings pinpointed several optimization targets as highlighted by red arrows in Fig. 2a: (i) the accessibility of the competing, unproductive P-twist channel should be minimized such that the excited-state wavepacket proceeds along the desired I-twist pathway; (ii) the efficiency of internal conversion as well as (iii) the photoreactivity at the I-twisted intersection seam need to be increased to promote efficient photoisomerization.

Like many monomethine dyes, HBDI$^-$ features torsion-coupled intramolecular charge transfer[22–24]. In particular, the two torsional modes are associated with oppositely directed charge transfer[25–27]. This immediately presents an avenue for manipulating the decay pathways through preferential charge-state stabilization. Based on a three-state state-averaged complete active space self-consistent field (SA3-CASSCF) treatment of the electronic structure, Olsen and McKenzie presented a three-state diabatic representation that captures this charge-localized picture (Fig. 2b, c)[28]. In this model, the pertinent fragment charge-localized diabatic states (|P>, |I> and |B> labeled according to their dominant electron localization on the P-ring, I-ring or methine bridge, respectively) are constructed from the effective covalent Hamiltonian obtained by block-diagonalizing[29,30] the full Hamiltonian in the basis of singlet configuration-state functions into covalent and ionic blocks (Supplementary Figs. 2 and 3). In this framework, engineering the potential energy landscapes boils down to modifying the relative energetics of the charge-localized diabatic states (also called the driving force) and their electronic couplings (Fig. 2b).

Chemical modifications of the chromophore offer molecular knobs to manipulate the diabatic-state properties. There have been numerous studies on effects of introducing substituents on the I-ring[31–34], as well as of smaller perturbations on the P-ring (including alkylation, methoxylation, and halogenation)[35–39]. These have focused on vertical properties (e.g., emission color, electron detachment energy, excited-state acidity) and timescales of fluorescence decay. Here, we target selected chemical P-ring modifications to investigate their impact on internal conversion pathways and specifically explore diabatic-state biasing as a potential route to steer excited-state dynamics and overcome photoisomerization bottlenecks in HBDI$^-$. We consider the 2,3,5-F$_3$ (TFHBDI$^-$) and 3-OCH$_3$ (MHBDI$^-$) substituted chromophores (Fig. 1) that are characterized by opposite electronic character (overall σ-electron-withdrawing and π-electron-donating[40], respectively). Our choice was inspired by recent experimental work by Boxer and coworkers on effects of electron-donating and electron-withdrawing P-ring substituents on the fluorescence quantum yield (FQY) of Dronpa2 and superfolder GFP[41]. They reported a decrease in the FQY and estimated excited-state torsional barrier as a function of transition energy irrespective of the electronic nature of the substituent (except for the heavier halogens)[41]. In particular, both 2,3,5-trifluorination and 3-methoxylation resulted in a drop in the FQY in the Dronpa2 scaffold from 47% in the wildtype to 13 and 36%, respectively. A smaller but similarly directed effect (from 57% to 43 and 11%, respectively) was found for the corresponding

superfolder GFP variants[41]. This peaked trend in FQY (Fig. 3a in their work[41]) led the authors to propose that the dominant torsional deactivation mechanism depends on the electronic character of the substituent. However, fluorescence does not report on the underlying decay pathways and hence, the mechanistic details remain elusive.

Here, we use simulations to investigate the possibility of selectively controlling excited-state pathways and reactivity via P-ring substitution. We focus on the isolated setting, which allows us to distinguish direct substituent effects (potential/electronic and inertial[42]) on the chromophore from derived effects caused by modified steric and electrostatic interactions with a surrounding environment. Our results suggest electron-withdrawing P-ring engineering as an effective route to promote selective I-twisting and photoisomerization through synergistic potential and inertial substituent effects. This detailed insight is a key component to advance our ability to engineer chromophore excited-state reactivity and represents a stepping stone toward photofunctional design in the more complex setting of a protein scaffold.

## Results and discussion

**Static picture of substituent effects.** We first investigate the electronic effects of the P-ring substituents by comparing the relative ground- and excited-state energies of TFHBDI$^-$ and MHBDI$^-$ at key critical geometries to their unmodified counterparts (Fig. 3). These calculations were performed using three-state extended multistate second-order perturbation theory with an active space consisting of four electrons in three π-orbitals (Supplementary Fig. 2) together with the 6-31 G* basis set (SA3-XMS-CASPT2(4,3)/6-31 G*). The more efficient empirically-corrected complete active space self-consistent field (α-CASSCF) method was used in the three-state diabatic analysis and in the nonadiabatic dynamics simulations (Supplementary Note 1). Geometric parameters of critical points and associated energies are reported in Supplementary Fig. 4 and Supplementary Tables 1–4/6–13.

At the FC point, the P-ring substituents impart only small effects: 3-methoxylation induces a red-shift (~0.07 eV) and 2,3,5-trifluorination a blue-shift (<0.05 eV) (Supplementary Table 5). According to the three-state diabatic analysis (Supplementary Note 1, Supplementary Fig. 5 and Supplementary Tables 16/17), the substituents in all cases increase the energy difference (driving force) between the pertinent |P> and |I> diabatic states (HBDI$^-$: −0.038 eV; TFHBDI$^-$: −0.198 eV; MHBDI$^-$: −0.082 eV). This suggests a blue shift in both cases. However, the coupling between the |P> and |I> diabatic states is increased in TFHBDI$^-$ (1.112 eV compared to 1.092 eV in HBDI$^-$) while it is reduced in MHBDI$^-$ (1.065 eV). For MHBDI$^-$, this effect counteracts the increase in the driving force, thereby accounting for the observed red-shift. This follows from the fact that the |P> and |I> diabatic states have comparable energies at planar geometries, which implies that the S$_1$/S$_0$ energy gap is largely dictated by the electronic couplings (planar (FC), Fig. 2b, c). In other words, the main factor underlying the substituent effects at the FC point is their small modulation of the electronic couplings rather than the driving force. Interestingly, this picture is opposite of that proposed by Lin et al. for environmental effects on photophysical properties of GFPs[43]. They found that the influence of protein mutations could be strongly correlated with changes in the driving force, assuming a fixed coupling strength within a Marcus–Hush framework. This difference may not be too surprising given the much stronger electronic effects of chemical substituents on the chromophore compared to those of mutations in the surrounding scaffold. However, as we will discuss next, changes in the driving force

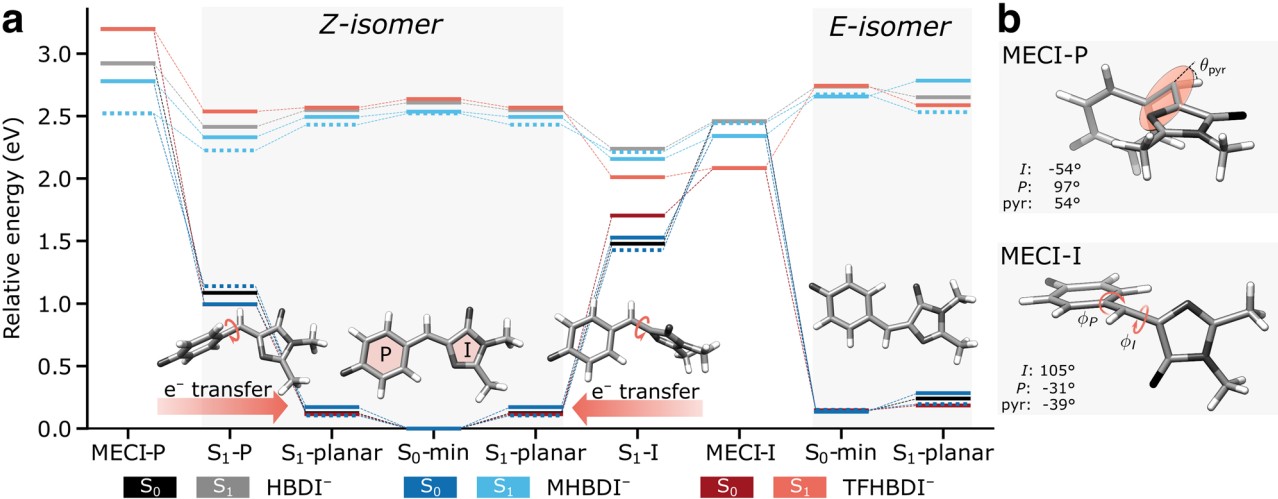

**Fig. 3 Substituent effects on critical points on the potential energy surfaces important for the excited-state decay. a** Ground- and excited-state potential energy surfaces at minima and MECIs for HBDI⁻, MHBDI⁻ and TFHBDI⁻ as computed at the SA3-XMS-CASPT2(4,3)/6-31 G* level of theory. For HBDI⁻, the MECI-I corresponds to MECI-I⁺ (see Fig. 7). 3-methoxylation leads to either a slight (de)stabilization of the (I)P-twisted geometries (in-plane methoxy orientation, dashed blue lines) or essentially no change (out-of-plane, solid blue lines) in relative energetics. Insets in Fig. 4 illustrate the in-plane/out-of-plane orientations. On the other hand, 2,3,5-trifluorination leads to a stabilization of the I-twisted geometries and destabilization of the P-twisted counterparts. This brings the MECI-I structure in proximity of the twisted S₁-I minimum. Note that the asymmetric substitution patterns in MHBDI⁻ and TFHBDI⁻ break the symmetry between the oppositely directed P- and I-twisted MECIs in HBDI⁻. However, since the energy differences are small, we only report one for each torsional mode. **b** MECI structures for HBDI⁻ highlighting the key bridge torsion and pyramidalization parameters (see also Supplementary Fig. 1).

become the dominant substituent effect at twisted structures where the |P⟩ and |I⟩ states are no longer electronically coupled.

While the substituent effects are modest at the FC point, they become pronounced upon bridge torsion. This is a result of the concomitant decoupling of the |I⟩ and |P⟩ states (vanishing off-diagonal element, see Fig. 2b), which means that the dominant substituent effects enter the energies (zeroth order) rather than through the electronic couplings (i.e., first order). The overall σ-electron-withdrawing nature of the fluorine atoms increases the electron affinity of the P-ring, thereby stabilizing |P⟩ and de-stabilizing |I⟩. This explains the decreased S₁/S₀ energy gap at I-twisted geometries in TFHBDI⁻ where the ground state is dominated by the |I⟩ state, and the other way around for P-twisted configurations. Because of this reduced gap, the minimum on the I-twisted intersection seam can be reached without significant methine bridge pyramidalization (~18°/0.4° for XMS-CASPT2/α-CASSCF in TFHBDI⁻ relative to ~39°/30° in HBDI⁻, where 55° corresponds to an idealized sp³ hybridized C-atom). Besides this, the main difference between the S₁-I minimum and MECI-I amounts to small variations in bond-length alternation (BLA) towards a slightly increased contribution of the phenolate resonance structure at MECI-I. As a result, MECI-I is energetically close to the twisted minimum (within <0.1 eV). On the other hand, the diabatic-biasing effect of trifluorination removes the driving force along the P-torsional mode and substantially increases the uphill path toward the P-twisted intersection seam, locating MECI-P more than 0.5 eV above the FC point. For MHBDI⁻, the extent of the potential effects depends on the methoxy group orientation (see insets in Fig. 4, Supplementary Note 2 and Supplementary Fig. 6) due to the change in hybridization of the oxygen atom. For the in-plane orientation (dashed blue lines, oxygen is sp²-hybridized), the overall π-electron-donating character of the methoxy group stabilizes and destabilizes the |I⟩ and |P⟩ states, respectively. This leads to opposite diabatic-biasing trends for the two twisted S₁ minima relative to TFHBDI⁻, leaving them essentially isoener-getic. On the other hand, the ground-state favored out-of-plane

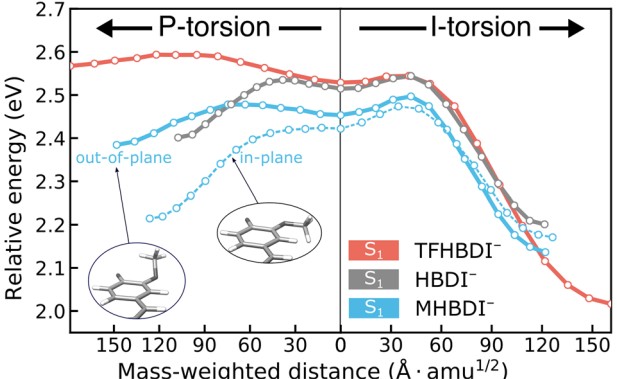

**Fig. 4 Substituent effects on excited-state torsional barriers.** Relative XMS-CASPT2(4,3)/6-31 G* energies along the S₁ minimum energy path connecting the S₁-planar geometry to each of the two twisted minima (left: S₁-P, right: S₁-I). The path was computed using NEB at the α-CASSCF(4,3)/6-31 G* level and hence the end points do not exactly match those in Fig. 3. While the barrier estimates for I- and P-twisting are approximately the same in HBDI⁻ (gray), leading to almost equal bifurcation along the two competing pathways, both potential and inertial effects of 2,3,5-trifluorination (orange) act in concert to remove the driving force along the P-torsion in TFHBDI⁻. The effect of 3-methoxylation (blue) depends on the orientation of the methoxy substituent. In the ground-state favored out-of-plane orientation (solid line), the relative energetics of the twisted S₁ minima and barrier estimates resemble those of HBDI⁻. In the in-plane orientation, the two twisted S₁ minima become nearly isoenergetic with a small barrier of (~0.05 eV) along the I-torsion.

orientation (solid blue lines, increasingly sp³-hybridized) barely changes the relative energetics with respect to HBDI⁻.

To examine possible bottlenecks along the torsional modes, we further considered the influence of the substituents on the excited-state torsional barriers. Figure 4 shows estimates for XMS-CASPT2 S₁ barriers (in mass-weighted coordinates)

between the planar $S_1$ geometry and the twisted minima as computed based on nudged-elastic-band (NEB[44]) paths obtained at the α(0.64)-CASSCF(4,3)/6-31 G* level of theory. As such, these values represent upper bounds to true barriers. The inertial and potential effects are oppositely directed along the I-torsion in TFHBDI⁻, with the inertial slowing due to the heavier fluorine substituents being counteracted by the steeper gradient along the I-torsional mode. Conversely, both inertial and potential effects act in concert to remove the driving force along the P-torsion. In MHBDI⁻, the details of the potential effects depend on the orientation of the methoxy group (out-of-plane/in-plane relative to the P-ring) but the overall picture is the following: the I-twist pathway may be associated with a small barrier (~0.05 eV) while the P-twist pathway has at most a similarly small barrier (and may be barrierless). However, these potential effects are, at least partly, counteracted by the heavier mass of the methoxylated P-ring. As such, our static results suggest that electron-withdrawing groups on the P-ring indeed promote the desired I-torsional pathway through synergistic potential and inertial effects, whereas electron-donating P-ring substituents lead to opposing effects and, therefore, do not efficiently promote selective P-twisting. Motivated by its desired bias toward the I-torsional mode, we proceeded to explore the dynamical implications of 2,3,5-trifluorination on the photoisomerization quantum yield.

**Bond-selectivity and accelerated internal conversion**. To investigate how trifluorination influences the internal conversion behavior, we performed AIMS simulations at the α(0.64)-SA3-CASSCF(4,3)/6-31 G* level of theory. The resulting $S_1$ population decay profile for gas-phase TFHBDI⁻ is shown in Fig. 5 alongside that of HBDI⁻ from our previous work[15]. The HBDI– decay can be fitted to a delayed biexponential decay that gives three time scales; a fast (~180 fs), an intermediate (~1 ps) and a longer-lived (~10 ps) component, in line with experiments[45,46]. We attributed these to the departure from the FC point, internal conversion through the I-twist pathway and trapping on $S_1$ along the P-torsion. Trifluorination has a profound effect on these time

scales. While the lag time is only slightly increased to ~200 fs, the internal conversion is now significantly accelerated with the intermediate decay constant of 361 ± 56 fs, which is more than halved compared to HBDI⁻ (909 ± 170 fs). To understand this difference, we examined the $S_1$ reduced density along the bridge torsional modes. As seen by comparing Fig. 6, trifluorination exerts two main effects: (i) it significantly enhances the population transfer along the I-torsional mode (~70% of the population is transferred during the first approach to the intersection seam, compared to ~25% in HBDI⁻), removing the oscillations around the I-twisted minimum (compare Fig. 6a/c); and (ii) it effectively turns off the unproductive P-twist channel (compare Fig. 6b/d) with less than 5% of the population undergoing initial P-twisting (see also Supplementary Fig. 7/8). Note that this small P-twisted subpopulation is likely a result of the overstabilization of the twisted structures in the α-CASSCF-based dynamics (see Supplementary Fig. 4).

To understand the observed difference in population transfer efficiency, we considered the dynamically accessed regions of the I-twisted intersection seam. Figure 7a compares the distributions of the nonadiabatic transition events along the I-torsion and methine HOOP (pyramidalization) modes for HBDI⁻ and TFHBDI⁻ together with the associated potential energy curves that indicate Z- and E-isomer wells (Fig. 7a, b). These coordinates contribute to the first-order branching space (spanned by the gradient-difference vector: g-vector and the derivative coupling vector: h-vector), which defines the MECI-Is (Supplementary Fig. 9). The h-vector mainly represents I-torsion with the +h-direction corresponding to torsional motion toward the E-isomer. The g-vector corresponds to bond-length alternation (in HBDI⁻, it also involves pyramidalization). In our previous work on HBDI⁻[15], we found a bimodal distribution of nonadiabatic transfers around the I-twisted intersection seam that was caused by the presence of two near-symmetry related MECI-Is (denoted MECI-I⁺ and MECI-I2⁺, indicated by yellow diamonds in Fig. 7a, b with the connecting seam minimum energy path (MEP) shown as a line). These correspond to I-torsional angles slightly above and below 90°, respectively. Upon trifluorination, these peaks effectively merge into a unimodal distribution (with its center located slightly below 90° along the I-torsion and at slightly

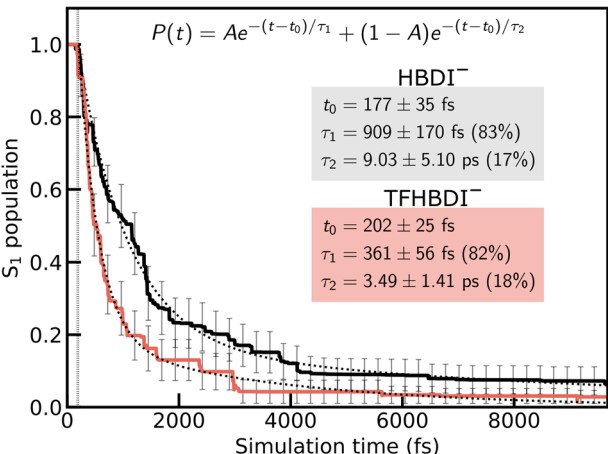

**Fig. 5 Effects of trifluorination on the excited-state decay constants.** $S_1$ population decay for HBDI⁻ (black line) and TFHBDI⁻ (red line) obtained from the α(0.64)-SA3-CASSCF(4,3) AIMS dynamics together with the biexponential fits (dashed lines). The inset labels give the lag time ($t_0$) as well as the time constants of the fitted decays. Errors estimates are based on bootstrapping with 1500 boot cycles applied to the population traces of each initial condition. In TFHBDI⁻, 80% of the $S_1$ population is depleted within ~1 ps while less than 5% remains trapped on $S_1$ by the end of the simulation time (~10 ps).

$$P(t) = A e^{-(t-t_0)/\tau_1} + (1-A) e^{-(t-t_0)/\tau_2}$$

HBDI⁻
$t_0 = 177 \pm 35$ fs
$\tau_1 = 909 \pm 170$ fs (83%)
$\tau_2 = 9.03 \pm 5.10$ ps (17%)

TFHBDI⁻
$t_0 = 202 \pm 25$ fs
$\tau_1 = 361 \pm 56$ fs (82%)
$\tau_2 = 3.49 \pm 1.41$ ps (18%)

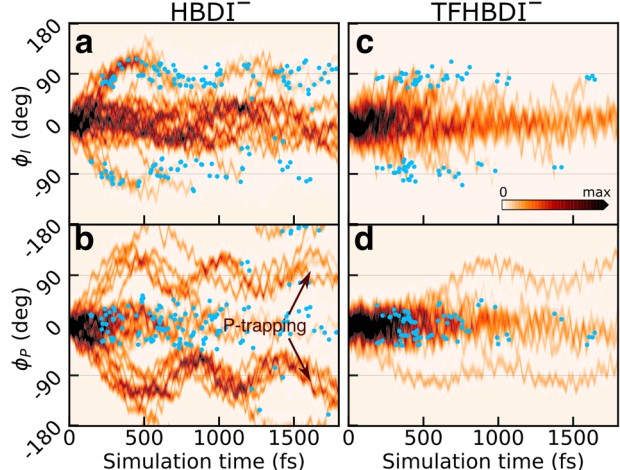

**Fig. 6 Effects of trifluorination on the excited-state deactivation mechanism.** Time evolution of the reduced $S_1$ density along the $\phi_I$ and $\phi_P$ torsional modes within the first 1.8 ps after photoexcitation: (**a**, **b**) HBDI⁻ and (**c**, **d**) TFHBDI⁻. Blue filled circles indicate population (nonadiabatic) transfer events. While HBDI⁻ proceeds along both P- and I-torsional modes, TFHBDI⁻ undergoes I-twisting almost exclusively. The reduced densities were computed using a Monte Carlo procedure[62].

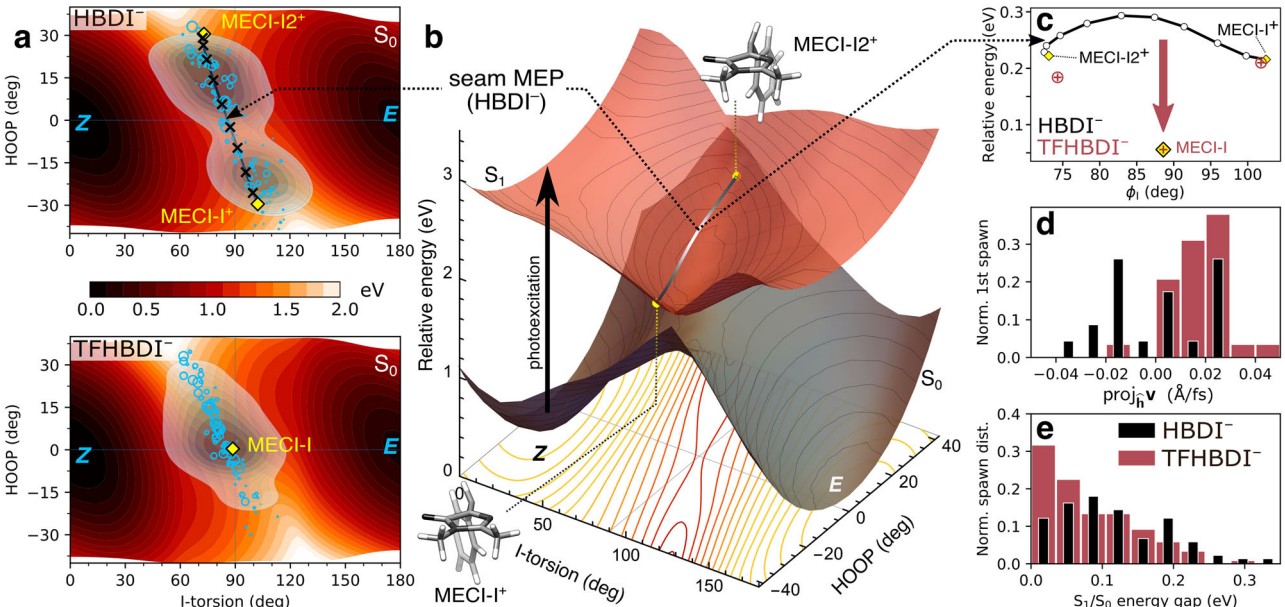

**Fig. 7 Effects of trifluorination on the approach to the positive I-twisted intersection seam. a** Contour plots of the $S_0$ PES along the I-torsion and HOOP modes in HBDI⁻ (top) and TFHBDI⁻ (bottom). These were obtained by an unrelaxed HOOP scan along a scan of the I-torsion keeping the P-torsion fixed at zero, while all remaining coordinates were allowed to relax as described in Section S6 of our previous work[15]. MECIs are highlighted by the yellow diamonds, and the gray line connecting MECI-I⁺ and MECI-I2⁺ for HBDI⁻ indicates the seam MEP. Non-adiabatic transitions are indicated by blue open circles with area scaled according to the absolute population transfer. The associated distributions, obtained by convolution with Gaussian functions, are shown as white-to-blue contours. The approximate bimodal distribution in HBDI⁻ becomes unimodal upon trifluorination. **b** Three-dimensional representation of the $S_0$ and $S_1$ PESs in HBDI⁻ where the contour plot below shows the energy gap. The MECIs are shown as yellow points, and the seam MEP as the connecting gray line. **c** The seam MEP for HBDI⁻, connecting MECI-I⁺ and MECI-I2⁺. The analogue structures in TFHBDI⁻ were computed as minimum-distance conical intersections using the trifluorinated HBDI⁻ structures as reference geometries. These are shown as red circles with a plus together with the MECI-I (yellow diamond with a plus). **d** Normalized distributions of velocity components for the parent TBF along the $h$-direction (dominated by I-torsional motion, see Supplementary Fig. 9) for the earliest non-adiabatic transition for each initial condition events along the I-twisted intersection seam. **e** Normalized distributions of $S_1/S_0$ energy gaps for the non-adiabatic transition events along the I-twisted intersection seam. Events for both positive and negative $\phi_I$ directions have been combined.

positive pyramidalization angle). As expected from the MECIs, torsional motion is enough to bring the $S_0$ and $S_1$ surfaces for TFHBDI⁻ into near-degeneracy and facilitate nonadiabatic transitions. Accordingly, the population transfer is no longer gated by the HOOP motion. As shown in Fig. 7c, this is a result of that trifluorination effectively turns the transition state (~0.1 eV barrier) on the seam MEP between MECI-I⁺ and MECI-I2⁺ into a minimum (~0.15 eV lower in energy and within 0.1 eV of the $S_1$ minimum) as indicated by the vertical arrow in Fig. 7c. While the MECI-Is in both HBDI⁻ and TFHBDI⁻ have similar sloped topographies, the larger transfer efficiency upon trifluorination is a consequence of two effects. First, the modification of the intersection seam leads to a more direct (ballistic) approach to the seam that increases the velocity component along the direction of the derivative coupling vector ($h$-vector), and in turn, the effective coupling strength (Fig. 7d). The implications of the almost exclusive component along the $+h$-direction in TFHBDI⁻ will be discussed further below. Indeed, as discussed by Malhado and Hynes[47], a sloped intersection may be as efficient as a peaked one to mediate transfer provided the wavepacket is steered towards the intersection seam. Second, the narrowing of the $S_1/S_0$-energy gap at twisted geometries upon trifluorination extends the effective transfer region around the seam (Fig. 7e).

**Photoisomerization quantum yield.** A pertinent question to ask is to what extent the more ballistic progress along the isomerization-driving I-twist coordinate induced by trifluorination affects the outcome of the internal conversion (i.e., successful or aborted isomerization around the I-bond). To estimate the

photoisomerization quantum yield, we followed the ensuing $S_0$ dynamics for 150 fs after the nonadiabatic transition events and classified the stereoisomer distribution (within ±55° relative to the Z- and E-isomer). Our previous study of HBDI⁻ demonstrated that this propagation time was sufficient to identify the photo-product (i.e., longer $S_0$ propagation times did not affect the photoproduct assignment)[15]. For TFHBDI⁻, we find that a total of 58% of the excited-state population undergoes photo-isomerization to the E-isomer, corresponding to an almost doubling relative to HBDI⁻ (~30%).

To understand the improved photoreactivity of the trifluorinated chromophore, we performed so-called cone sampling and photo-isomerization committor analysis[15] (Supplementary Methods 2). In short, we sample configurations within the branching plane (spanned by the $g$- and $h$-vectors) of each MECI-I and then perform 300-fs dynamics on the ground state for each of these samples. The ground state dynamics is done in three ways (corresponding to three distinct limiting behaviors): (i) the photoreactivity imprinted in the ground-state potential around the intersections as given by the minimum energy paths on $S_0$. This corresponds to the artificial situation where all kinetic energy is quenched after each infinitesimal time step during ground-state dynamics; (ii) starting from zero initial velocities but now accounting for the inertia gained on the ground state; and (iii) similar to case (ii) but now starting from 50 randomly sampled initial velocities for each configuration. This provides an estimate for the probability of photoproduct generation under the assumption of energy equipartitioning between all nuclear degrees of freedom. While thermalization is not expected within the

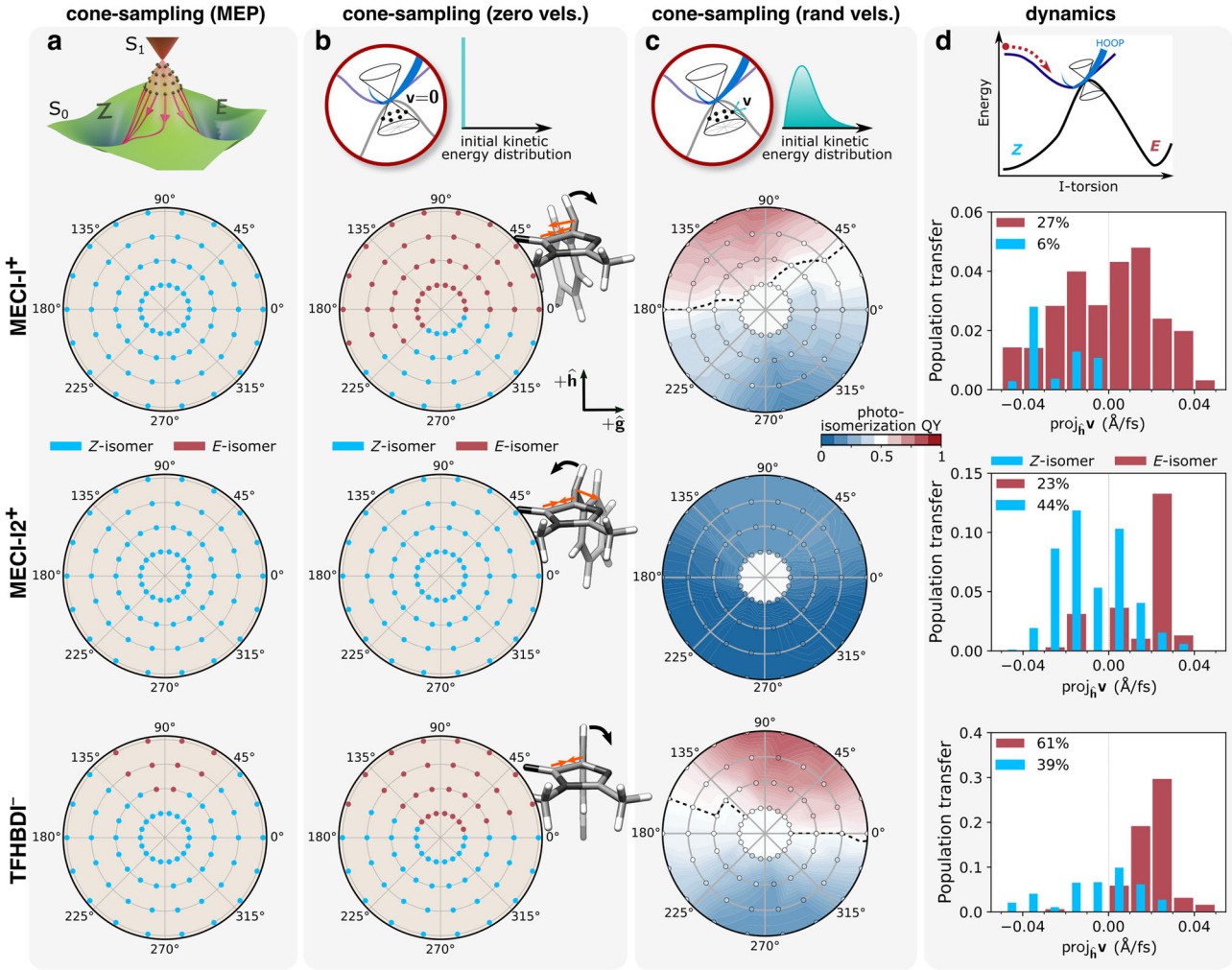

**Fig. 8 Effects of trifluorination on the photoreactivity of the I-twisted intersection seam.** First row: Schematic of the three sampling schemes; second and third rows: MECI-I+ and MECI-I2+ for HBDI−; fourth row: MECI-I for TFHBDI−. Photoproduct distribution at each displacement within the branching plane (for details, see Supplementary Methods 2), based on the outcome of dynamics starting from (**a**) zero initial velocities and quenching the kinetic energy at every infinitesimal time step, i.e., corresponding to the minimum energy path (MEP); (**b**) zero initial velocities. Orange arrows indicate the force on the I-ring upon transfer to $S_0$, and the black arrow the dominant resulting motion; (**c**) 50 initial conditions with randomized velocities. The black line indicates the equicommittor line, indicating 50% photoproduct generation; (**d**) Distributions of the velocity components at the nonadiabatic transition events along the $h$-direction (dominated by I-torsional motion) for the parent TBFs on $S_1$ that follows the I-twist pathway, classified according to the outcome of the ground-state dynamics. Events for both positive and negative $\phi_I$ directions have been combined. Percentages are relative to the part of the population for which decay is mediated by I-torsion.

ultrafast timescales of the dynamics, this offers insight into whether inertial effects imprinted by the ground state potential survive the more realistic setting of having initial kinetic energy in all degrees of freedom. Figure 8 summarizes the results for the two near-symmetry-related MECI-Is in HBDI− and that in TFHBDI− in the form of polar plots. Here, the location of each point represents geometric displacement within the branching plane and its color indicates the outcome of the ground-state dynamics.

Figure 8a shows the outcome upon following the paths of steepest descent on $S_0$ (MEP). While the MECI-Is in HBDI− lead exclusively to recovery of the Z-isomer, larger displacements along the isomerization-driving +$h$-direction leads to E-isomer generation in TFHBDI−. In other words, photoreactivity is already imprinted in the ground-state PES near the MECI-I of TFHBDI−. This can be understood by the fact that $\phi_I \sim 90°$ such that even small displacements along the isomerization-driving coordinate (+$h$-direction) will promote photoisomerization. Figure 8b, c shows the results of the zero- and random-initial-velocity sampling schemes together with the results previously

obtained for the two near-symmetry-related MECI-Is in HBDI−[15]. Accounting for the inertia gained on the ground state leads to an increase in the photoreactivity of TFHBDI−. As in HBDI−, the direction of steepest descent involves a rapid asymmetric contraction of the I-ring (contraction of the $C_5$-$C_6$ bond, orange arrows in Fig. 8b) that promotes photoisomerization for configurations above 90°. However, the ground-state inertial effects are less pronounced in TFHBDI− relative to MECI-I+ in HBDI− because the methine bridge planarization, and the accompanying HOOP motion, is significantly reduced. As shown in Fig. 8c, the photoreactivity encoded in both potential and inertial effects on the ground state is largely preserved when injecting initial kinetic energy in all degrees of freedom. Here, the dashed black line represents the photoisomerization equicommittor that corresponds to the geometries at which initialization of $S_0$ dynamics leads to Z- and E-isomers with equal probability. The isomerization-promoting potential effects in TFHBDI− manifest in a generally increased photoisomerization probability (darker red shade) as compared to MECI-I+ in HBDI−.

Having considered these constructed regimes, it is critical to examine the actual, nonequilibrium velocity distribution at the intersection seam and its impact on the photoproduct generation. Figure 8d displays the distributions of velocities along the $h$-vector direction at the nonadiabatic transfer events with color coding according to the resulting photoproduct. This direction was chosen because it is dominated by the isomerization-driving I-torsional coordinate. As discussed in our previous work[15] and as evident here, there are two factors governing photoisomerization in HBDI⁻: (i) the symmetric velocity distribution of the photoproduct-generating nonadiabatic transfer events around the more reactive MECI-I+; and (ii) the skewed photoproduct-generating distribution around the less reactive MECI-I2+. This shows that photoisomerization in this region of the intersection seam occurs in the ballistic regime where the positive velocity component along the isomerization-driving direction prevails. It is important to note that the combined velocity distributions around either of the MECI-Is in HBDI⁻ are essentially symmetric. By contrast, it is highly skewed toward positive velocity components in TFHBDI⁻. This part of the nonadiabatic transfer events is further responsible for photoproduct generation. In other words, there are two factors contributing to the doubled photoisomerization quantum yield in TFHBDI⁻: (i) a more ballistic approach to the intersection seam along the isomerization-driving coordinate. This follows from the stabilization of the I-twisted intersection seam upon trifluorination, which brings its minimum in energetic proximity of the I-twisted $S_1$ minimum; and (ii) the accessed part of the intersection seam is both more reactive and its nonadiabatic transfer efficiency higher.

In summary, we have investigated the possibility to chemically control the internal conversion in the HBDI⁻ chromophore with focus on improving its photoisomerization propensity. We considered 3-methoxylation and 2,3,5-trifluorination of the P-ring to examine the influence of two electronically contrasting, perturbative modifications. At the FC point, the substituent effects are minor (absolute shifts of <0.1 eV): in the three-state diabatic framework of Olsen and McKenzie, the resulting shifts are dominated by small modulations of the electronic couplings between the diabatic states more so than changes in their relative energies. This picture is reversed at twisted structures where the electronic coupling between the I- and P-ring charge-localized states ceases, and diabatic-state biasing effects become dominant. The overall σ-electron-withdrawing fluorine substituents bias the excited-state potential toward I-twisting and essentially remove the energetic driving force along P-torsion, which otherwise characterizes the unmodified chromophore. This potential effect is further accentuated by a larger inertial influence of trifluorination on the P- compared to the I-torsional mode. The extent of the potential effects of 3-methoxylation depends on the orientation of the methoxy group and, hence, the hybridization of the methoxy oxygen. While the increasingly sp²-hybridized in-plane conformation does slightly promote P-twisting, it is only just enough to counteract the asymmetry inherent in HBDI⁻ caused by the electronegativity differences of the I- and P-rings. However, P-twist promoting potential effects will, at least partly, be counteracted by the inertial slowing upon P-ring substitution. As such, adding electron-donating P-ring substituents does not appear to be a successful strategy for selective P-twisting. On the other hand, introducing electron-withdrawing groups on the I-ring may lend itself ideal for this purpose, as indicated by previous work on the red $N$-acylimine I-ring substituted HBDI⁻ analogue[48].

Based on the desired I-twist favoring changes caused by 2,3,5-trifluorination, we further simulated its dynamical impact upon photoexcitation. The effects of trifluorination can be summarized in three key points: (i) it essentially turns off the unproductive P-twisting channel and hence increases the propensity of the system to progress along the potentially reactive pathway. This is a result of an increase in the effective torsional excited-state barrier through both potential and inertial effects; (ii) it accelerates internal conversion via the I-twisted pathway by changing the location of the minimum on the intersection seam to be near the twisted $S_1$ minimum. This changes the direction and velocity of approach to the I-twisted CI seam, which is no longer gated by the HOOP motion; (iii) as such, trifluorination induces a more ballistic behavior along the isomerization-driving coordinate toward the I-twisted CI seam which, in turn, almost doubles the photoisomerization quantum yield (from ~30 to ~60%).

The extent to which this design strategy is transferable to the confined environment of a protein requires further investigation. Beyond the gas phase, chemical substituents not only modify the intrinsic chromophore behavior but also its interactions with the surrounding environment through changes in sterics and electrostatics. For example, the phenolate and imidazolinone oxygens typically participate in hydrogen bonding with neighboring residues (e.g., Ser142 in Dronpa2), and the redistribution of electron density toward the electron-withdrawing fluorines may impact the hydrogen-bond-acceptor capacity of the phenolate oxygen, and hence the electrostatic environment. Furthermore, the larger displacement volume required for I-twisting may render the one-bond-flip centric picture (characterizing the gas-phase dynamics) dubious inside protein scaffolds. We hope our theoretical work will inspire complementary experimental efforts into quantifying the intrinsic photoisomerization propensity of HBDI⁻ derivatives (e.g., using a combination of isomer-selective experiments and time-resolved spectroscopy), and how it is affected upon embedding into the protein β-barrels.

## Methods

**Electronic-structure theory levels and geometry optimizations**. Initial investigations of the potential effects of 2,3,5-trifluorination and 3-methoxylation were performed at the extended multistate multireference second-order perturbation theory (XMS-CASPT2[49]) level. An active space consisting of four electrons in three orbitals (the bonding, non-bonding and anti-bonding methine bridge orbitals, see Supplementary Fig. 2) with state-averaging over the three lowest singlet states and the 6-31 G* basis set was used (frozen core; real level shift of 0.3 a.u.; SVP-jkfit density-fitting basis), i.e., SA3-XMS-CASPT2(4e,3o)/6-31 G*. XMS-CASPT2 calculations were performed using the BAGEL program[50,51].

Following our previous work on HBDI⁻[15], we employed the more efficient empirically-corrected α-complete active space self-consistent field (α-CASSCF[52]) method as electronic-structure engine (adiabatic energies, nuclear gradients and nonadiabatic couplings) in the nonadiabatic dynamics simulations as well as in the three-state diabatic analyses and cone sampling (Supplementary Methods 1 and 2). Active space, state-averaging and basis set used were the same as above. The fitting procedure for the α-parameter and the validation against XMS-CASPT2 at critical point geometries are described in Supplementary Note 1. All α-CASSCF calculations were performed using a development version of the TeraChem program[53–56]. Geometry optimization, minimum energy conical intersection (MECI) searches and minimum energy pathways (MEPs) at this level of theory were computed using the DL-FIND[57] geometry optimization library and seam MEPs using pyGSM[58,59], both interfaced with TeraChem.

**Adiabatic and nonadiabatic dynamics simulations**. Initial conditions (ICs) for ab initio multiple spawning (AIMS) simulations were sampled from a ground-state harmonic Wigner

distribution at 300 K, with normal modes and harmonic frequencies computed at the MP2/cc-pVDZ level of theory. To avoid artificially long C–H bonds, caused by the linearization of the methyl torsions in the harmonic approximation, normal modes dominated by such rotations were excluded from the sampling. Absorption spectra were generated on the basis of 500 samples using the excitation energies and oscillator strengths provided by α-SA3-CASSCF(4,3)/6-31 G*. The stick spectra were convolved with Gaussian lineshapes with a full-width at half maximum of 0.07 eV. Given the absence of an experimental absorption spectrum for TFHBDI⁻, we applied the same uniform shift of +0.16 eV required to match the experimental absorption maximum for HBDI⁻. Similarly, 31 ICs were randomly sampled under the constraint that their vertical excitation energy is located within the spectral window of the pump pulse (2.48±0.05 eV) used in previous time-resolved photoelectron experiments on anionic HBDI⁻ in the gas-phase[45,60]. Only the two lowest singlet states were included in the AIMS simulations and each IC was initiated on $S_1$ under the independent first-generation approximation[61], i.e., they are uncoupled and run independently from the beginning, and propagated using AIMS for 10 ps or until the $S_1$ population dropped below 0.01. The classical equations of motion were integrated with an adaptive time step of 20 a.u. which was reduced upon encounter of non-adiabatic coupling regions. A spawning threshold of 0.005 a.u. (scalar product between derivative coupling and velocity vectors) was applied, and the minimum population of a trajectory basis function (TBF) to allow spawning was set to 0.01. TBFs on $S_0$ which have negligible overlap with other TBFs for at least 5 fs were uncoupled and continued for additional 150 fs to determine photoproduct. Bootstrapping (1500 samples) was used to calculate errors of simulated decay time constants.

## Data availability

The authors declare that all the other data supporting the findings of this study are available within the article and its supplementary information files and from the corresponding authors upon request. Validation of α-CASSCF level of theory, relative energies and geometric parameters at critical point geometries, analysis of the intersection parameters for the I- and P-twisted MECIs and results of three-state diabatic analysis are provided in the supplementary information. https://doi.org/10.5281/zenodo.10449450: Critical point geometries at the XMS-CASPT2 and α-CASSCF levels, initial conditions for AIMS dynamics.

## Code availability

The results in this manuscript were computed using TeraChem (http://www.petachem.com), Bagel (https://nubakery.org), pyGSM (http://github.com/ZimmermanGroup/pyGSM), and FMS90 (https://github.com/mtzgroup/fms).

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

## Acknowledgements

This work was supported by the AMOS program of the U.S. Department of Energy, Office of Science, Basic Energy Sciences, Chemical Sciences, and Biosciences Division. N.H.L. acknowledges financial support from the Villum Foundation (Grant No. VKR023371) and start-up funding from the School of Engineering Sciences in Chemistry, Biology and Health (CBH) at KTH Royal Institute of Technology; C.M.J. from the NSF graduate research fellowship program. Parts of the computations were enabled by resources (SNIC 2022/5-527) provided by the Swedish National Infrastructure for Computing (SNIC), partially funded by the Swedish Research Council through the grant agreement no. 2018-05973. The authors thank Dean Lahana for help to converge the MDCIs and Dr. Matthew Romei, Dr. Chi-Yun Lin and Prof. Steven Boxer for valuable discussions.

## Author contributions

N.H.L. contributed to the data curation, formal analysis, investigation, methodology, funding acquisition, resources, visualization, and writing (original draft) of the presented work. C.M.J. contributed to data curation, investigation, and methodology. T.J.M. contributed to methodology, formal analysis, funding acquisition and resources. All authors contributed to the conceptualization of the project, review and editing of the manuscript.

## Competing interests

The authors declare no competing interests.
