## [Peer Review File · Communications Chemistry]

Reviewers' comments:

Reviewer #1 (Remarks to the Author):

The work by List et al. reports a computational study on the impact of chemical substitutions in the phenol ring on the intrinsic non-adiabatic dynamics of the GFP chromophore anion (HBDI) following photoexcitation to S1. The results obtained by the authors provide detailed insight into the ways by which the internal conversion dynamics along the two competing reactive and non-reactive branches can be controlled. The successful strategy for accelerating internal conversion through the reactive pathway and increasing the photoisomerization quantum yield is introduced via fluorination of HBDI.

This is a very thorough and well executed study performed using state-of-the-art computational methods, which certainly deserves publication. However, the “results and discussion” section is written at a level that might not be easily understood by a general readership of Communications Chemistry, with many references to earlier works (e.g., Fig.5b-e, Fig6.d-e, Fig.7d). It seems to be more appropriate for specialists. Some explanation on the way of analyzing the results of the internal conversion dynamics can be elaborated. For example, HOOP modes are never defined in the manuscript, and although their role is evident for computational chemists dealing with photoisomerization reactions, it might not be so for general chemists. The same is true for the coordinates of the branching plane (Fig. 7).

Also, the quantum chemistry method used in the present study, which is the empirically corrected SA(3)-CASSCF(4,3)/6-31G*, although being validated through the higher-level XMS-CASPT2(4,3)/6-31G* method, might not be enough to reproduce the correct topography of the potential energy surfaces of the S0 and S1 states over a wide range of geometric parameters pertinent to the photoisomerization mechanism. I wonder, if any XMS-CASPT2(4,3)/6-31G* AIMS calculations can be carried out at this stage?

Reviewer #2 (Remarks to the Author):

The paper describes quantum chemical calculations on the energetics, electronic structure and excited state dynamics of the GFP chromophore anion. In addition to its importance in bioimaging and optobiology this chromophore has become something of a test bed for excited state calculations. Consequently, a great deal of high quality information has been generated and the excited state structure and dynamics have been well characterised. In this paper the key question raised and addressed is how to use those methods and results to influence the outcomes of chemical reactions. A key first step is made by showing that electron donating and withdrawing substituents significantly modify the branching ratio of the two excited state torsional relaxation pathways. This is an important result which will stimulate new experiments. The work is well described and clearly presented and will be of interest to many in the photochemistry and photobiology communities. Thus, the paper certainly warrants publication, and there are only a few minor points for the authors to consider.

Legend to Fig 1 – consider ‘...anionic GFP chromophores studied.’

Legend to Fig 2 – please add a little more detail to specify the meaning of red/orange arrows and

dash/solid lines.

Page 4 – four lines from bottom, is 'estimates' needed here - it conflicts with 'reported'?

Page 5 top, after FQY add 'as a function of transition energy'.

Legend to Fig 3, there is an (a) but no (b), so drop (a)?

In discussion (p11 and elsewhere) 'derivative coupling vector' and 'h-vector' are introduced but not explicitly explained in the context of CoIs. Consider adding a sentence and relevant references?

Legend to Fig 6 – the contours are not obviously blue.

P17, para 2, topic (ii). It is not clear that these three sentences are all concerning the same substituent effect under (ii) - can the connection be made clearer or section rewritten?

The discussion mentions specific H-bond structures which may have an impact when the chromophore is in the protein. It would be helpful to mention and speculate on the effects of nonspecific 'solvation dynamics' in polar/nonpolar fluid solvents? These are typically driven by the polar solvent response to permanent dipole moment (and others charge distribution) changes on excitation. How might these effects modify relaxation through one or other pathway. Such polar solvent dynamics seem to have only a modest effect on HBDI photophysics.

Finally, it would be very interesting to extend these studies to the neutral chromophore, for which a moderately efficient E-to-Z isomerization is the key step in reversibly photoswitchable proteins, and which generally has much richer photochemistry (including photoacid response) than the anion.

Reviewer #3 (Remarks to the Author):

This work investigates the isomerisation of GFP chromophores using multireference methods. Three different chromophores are studied in their anionic form in vacuo. The canonical chromophore was compared to two chemically modified HBDIs that were experimentally introduced by Boxer and coworkers. The main question is which of the two potential double bonds is selected for isomerisation and how the chemical modification will influence the selection.

The work is extensive (conical intersection optimization, minimum energy path and nonadiabatic dynamics) and done at the highest possible level. However, I found some portions of the text a bit difficult to follow and had to re-read them several times. The manuscript should be accepted after some minor revision which is mainly for clarification:

1) A reduced active space was established for the canonical GFP chromophore. Can the authors explain why it is still sufficient for the modified chromophores especially for the fluorinated version which is

involved in pi-electron donation? The methoxy substituted version can also extend the pi-conjugation of the aromatic ring via the oxygen.

2) The minimum energy pathways were computed for the in-plane and out-of-plane orientation of the methoxy groups separately. Is it possible for the orientation to change during the photoreaction? Was this observed during NAMD?

3) Recent studies on related chromophores have suggested the possibility of a combined isomerization of the single and double bond. Was this observed in NAMD simulations?

4) There is an inconsistency in Fi. 4. It looks like the S1-planar geometry of TFHBDI- is lower in energy than the S1-P minimum, but in fig. 3 its higher. Is the level of theory the same?

Response to Reviewer Comments

Reviewer 1:

The work by List et al. reports a computational study on the impact of chemical substitutions in the phenol ring on the intrinsic non-adiabatic dynamics of the GFP chromophore anion (HB DI^-) following photoexcitation to S $_1$. The results obtained by the authors provide detailed insight into the ways by which the internal conversion dynamics along the two competing reactive and non-reactive branches can be controlled. The successful strategy for accelerating internal conversion through the reactive pathway and increasing the photoisomerization quantum yield is introduced via fluorination of HB DI^- . This is a very thorough and well executed study performed using state-of-the-art computational methods, which certainly deserves publication.

However, the “results and discussion” section is written at a level that might not be easily understood by a general readership of Communications Chemistry, with many references to earlier works (e.g., Fig.5b-e, Fig6.d-e, Fig.7d). It seems to be more appropriate for specialists. Some explanation on the way of analyzing the results of the internal conversion dynamics can be elaborated. For example, HOOP modes are never defined in the manuscript, and although their role is evident for computational chemists dealing with photoisomerization reactions, it might not be so for general chemists. The same is true for the coordinates of the branching plane (Fig. 7).

We thank the reviewer for the overall positive feedback as well as the suggestions to strengthen the manuscript’s impact by improving its accessibility and readability. As detailed below, we have made changes to the text and figures to address the points raised by the reviewer.

Main modifications in the revised version of the manuscript:

- We have merged the original Figures 1 and 2 to a new Figure 1 (see below) that also includes schematics of the diabatic states and the diabatic Hamiltonian (subfigures c and d). These provide a visual representation of the diabatic-states and their energies at different geometries as well as the structure of the diabatic Hamiltonian while specialist details on the decomposition into underlying covalent and ionic states remains in the SI (Figure S6). This helps convey the diabatic state picture used to explain substituent effects in the Results section.
- We have clarified the content of former Figure 2 (new Figure 1b) by improving the caption, the surrounding text and by toning down the original HB DI^- behavior to emphasize the changes that are desired upon chemical substitution. We have introduced a “target changes” legend to further help the reading of this Figure.
- In the introduction, we now explicitly describe the HOOP mode when describing the excited-state behavior of HB DI^- (connecting to new Figure 1b).

- To improve the flow and focus of the main text, we have moved less important details in the “Results” section to the SI. Specifically, we have moved the discussion around the difluoro analogue to the SI. This aspect is an interesting check for the specialist but not relevant for the main message of the paper nor for the general reader.
- In the “Results” section, we now include a short summary of the excited-state dynamics of HBDI⁻ to provide background for the general reader about the unmodified system before reporting the TFHBDI⁻ results.
- We have included a short description of the branching space vectors to facilitate the discussion around our investigation of substituent effects on photoreactivity. This is intended to provide the necessary background for the general reader to appreciate the cone sampling analysis.

New Figure 1:

Fig. 1. (a) Chemical structures of the anionic GFP chromophores studied: the unmodified chromophore (HBDI⁻) together with its 3-methoxylated (MHBDI⁻) and 2,3,5-trifluorinated (TFHBDI⁻) derivatives. The pertinent P- and I-bonds are indicated by the arrows. (b) Schematic of modifications of the internal conversion dynamics of HBDI⁻ required to promote photoisomerization. Dashed lines represent the ground- and excited-state potential energy curves along the bridge torsions in HBDI⁻ (pyramidalization shown as blue curved lines) while solid lines

represent the desired modification with the targeted changes indicated by the orange arrows. In essence, the targeted effects are: (i) a bond-selective departure from the FC point along the reactive I-twist channel; (ii) a more direct approach to the I-twisted intersection seam to increase the internal conversion efficiency and, in turn, (iii) increase the photoisomerization quantum yield (thick, dark red arrows). Reaching the conical intersection seams (cones) in HBDI⁻ requires pyramidalization of the methine bridge, which is energetically uphill relative to the twisted minima. (c) Diabatic state energies (i.e., the diagonal elements of the effective Hamiltonians in (d)), their charge distribution and bonding character across the bridge. (d) Schematic of the three-state diabatic Hamiltonian which upon diagonalization gives the adiabatic states in (b). Progress along the bridge torsions leads to block-diagonal structures. The colored shading indicates the relative sign and magnitude of the matrix elements (diagonal: diabatic-state energies, off-diagonal: diabatic-state couplings). In the diabatic picture, substituents can affect the driving force (energy difference between diabatic states) and diabatic state couplings.

Also, the quantum chemistry method used in the present study, which is the empirically corrected SA(3)-CASSCF(4,3)/6-31G*, although being validated through the higher-level XMS-CASPT2(4,3)/6-31G* method, might not be enough to reproduce the correct topography of the potential energy surfaces of the S₀ and S₁ states over a wide range of geometric parameters pertinent to the photoisomerization mechanism. I wonder, if any XMS-CASPT2(4,3)/6-31G* AIMS calculations can be carried out at this stage?

We have in previous works (List, Martínez et al. Chem Sci. 2020, 11, 4180 and unpublished) interfaced AIMS with XMS-CASPT2 (Bagel and OpenMolcas), in principle enabling AIMS dynamics studies at the higher level of theory. However, the current interfaces do not leverage the possible parallelization in the AIMS ansatz with respect to trajectory basis functions (TBFs), which means that the simulation progress slows down significantly (formally N²) whenever new TBFs are spawned. We are working on overcoming this technical bottleneck. However, even with this solved, the computational scaling of XMS-CASPT2(4,3) limits the time scales (here covering the picosecond timescale) and ensemble averaging that can be achieved and hence the extent of the mapping of the dynamical effects studied in this work.

Our benchmark calculations show that α -CASSCF(4,3) provides a reasonably accurate yet computationally feasible approach to investigate these effects for the considered HBDI⁻ derivatives. Specifically, while the α -CASSCF(4,3) surfaces do not perfectly overlap the XMS-CASPT2(4,3) counterparts (shown at critical points in Figure S3) – the main difference being an overstabilization of the twisted configurations – the relative energetics of the two competing pathways is reproduced and so is the energy difference between each twisted S₁ minimum and the minimum on the respective conical intersection seam. This means that the wavepacket will have 0.2-0.3 eV more kinetic energy at the twisted minima (3-4 meV additional energy per degree of freedom under the

assumption of energy equipartitioning) at α -CASSCF(4,3) compared to XMSPT2. While this may affect the photoisomerization quantum yield to some degree, we expect the relative trends between HBDI⁻ and TFHBDI⁻ (i.e., the effect of trifluorination) to be retained given the similar overstabilization in the two systems. For these reasons, we decided to use the α -CASSCF(4,3) level in this study but agree that it would be interesting to pursue higher levels of theory in future work.

Reviewer 2:

The paper describes quantum chemical calculations on the energetics, electronic structure and excited state dynamics of the GFP chromophore anion. In addition to its importance in bioimaging and optobiology this chromophore has become something of a test bed for excited state calculations. Consequently, a great deal of high quality information has been generated and the excited state structure and dynamics have been well characterised. In this paper the key question raised and addressed is how to use those methods and results to influence the outcomes of chemical reactions. A key first step is made by showing that electron donating and withdrawing substituents significantly modify the branching ratio of the two excited state torsional relaxation pathways. This is an important result which will stimulate new experiments. The work is well described and clearly presented and will be of interest to many in the photochemistry and photobiology communities. Thus, the paper certainly warrants publication, and there are only a few minor points for the authors to consider.

We thank the reviewer for their positive comments on the manuscript, and detailed suggestions to improve the text and figures/legends.

Legend to Fig 1 – consider ‘...anionic GFP chromophores studied.’

Revised version:

Chemical structures of the anionic GFP chromophores studied: the unmodified chromophore (HBDI⁻) together with its 3-methoxylated (MHBDI⁻) and 2,3,5-trifluorinated (TFHBDI⁻) derivatives.

Legend to Fig 2 – please add a little more detail to specify the meaning of red/orange arrows and dash/solid lines.

As described in our response to Reviewer #1, we have made several changes to Figure 2, including merging it with Figure 1 (see subfigure b in new Figure 1 above). The modifications are outlined below.

- We have toned down the coloring (and changes the line style to dashed) of the unmodified chromophore (HBDI⁻) behavior to better highlight the desired changes (now as solid lines).
- We have added a “target icon” to help the understanding of the orange arrows as indicated the targeted substituent effects on the potential energy surface. The resulting dynamical effects of these potential energy effects are described in the caption and linked to the thick, red arrows in the figure.
- We have introduced numbering (i)-(iii) in the figure that directly links to the desired substituent effects discussed in the caption.

Page 4 – four lines from bottom, is 'estimates' needed here - it conflicts with 'reported'?

We used “estimates” in the original version of the manuscript because the excited-state barriers in the work by Romei *et al.* (Science, 2020, 367, 76) were estimates based on temperature-dependent fluorescence lifetime measurements (not a direct measurement). We changed the wording to clarify that the values reported by Romei *et al.* are estimated barrier heights.

Revised version:

They reported a decrease in the FQY and estimated excited-state torsional barriers as a function of transition energy irrespective of the electronic nature of the substituent (except for the heavier halogens).

Page 5 top, after FQY add ‘as a function of transition energy’.

We thank the reviewer for their detailed reading. We have included this missing detail in the sentence.

Revised version:

They reported a decrease in the FQY and estimated excited-state torsional barriers as a function of transition energy irrespective of the electronic nature of the substituent (except for the heavier halogens).

Legend to Fig 3, there is an (a) but no (b), so drop (a)?

In our version, we could not identify the missing (b) label in the original Figure 3 (new Figure 2). It is in the upper right corner, above the MECI-P structure. The subfigure b that shows the structures of the MECI-P and MECI-I structures is described in the last line of the caption.

In discussion (p11 and elsewhere) ‘derivative coupling vector’ and ‘h-vector’ are introduced but not explicitly explained in the context of CoIs. Consider adding a sentence and relevant references?

We thank the reviewer for pointing out this missing detail. In the revised version, we include a short description to explain.

Revised version:

Fig. 5a compares the distributions of the nonadiabatic transition events along the I-torsion and methine HOOP (pyramidalization) modes for HB BDI^- and TFHB BDI^- together with the associated potential energy curves that indicate *Z*- and *E*-isomer wells (Figs. 5a and b). These coordinates contribute to the first-order branching space (spanned by the gradient-difference vector: *g*-vector and the derivative coupling vector: *h*-vector), which defines the MECI-Is (Fig. S9). The *h*-vector mainly represents I-torsion with the +*h*-direction corresponding to torsional motion toward the *E*-isomer. The *g*-vector corresponds to bond-length alternation (in HB BDI^- , it also involves pyramidalization).

Legend to Fig 6 – the contours are not obviously blue.

We agree with the reviewer that the contour coloring is not clearly blue but rather goes from white-to-blue. In the new figure 5 (original Figure 6), we have corrected the caption to make the description more accurate and added thin, blue contour lines to better highlight the white-to-blue contours. We did not change the coloring of the contours themselves because this makes it more difficult to discern the nonadiabatic transition events (blue open circles).

New figure 5:

Fig. 2. Effect of trifluorination on the approach to the positive I-twisted intersection seam. (a) Contour plots of the S_0 PES along the I-torsion and HOOP modes in HB BDI^- (top) and TFHB BDI^- (bottom). These were obtained by an unrelaxed HOOP scan along a scan of the I-torsion keeping the P-torsion fixed at zero, while all remaining coordinates were allowed to relax as described in Section S6 of Ref. ¹⁵. MECIs are highlighted by the yellow

diamonds, and the gray line connecting MECI-I⁺ and MECI-I2⁺ for HBDI⁻ indicates the seam MEP. Non-adiabatic transitions are indicated by blue open circles with area scaled according to the absolute population transfer. ~~The associated distributions, obtained by convolution with Gaussian functions, are shown as white-to-blue contours.~~ The approximate bimodal distribution in HBDI⁻ becomes unimodal upon trifluorination. (b) Three-dimensional representation of the S₀ and S₁ PESs in HBDI⁻ where the contour plot below shows the energy gap. The MECIs are shown as yellow points, and the seam MEP as the connecting gray line. (c) The seam MEP for HBDI⁻, connecting MECI-I⁺ and MECI-I2⁺. The analogue structures in TFHBDI⁻ were computed as minimum-distance conical intersections using the trifluorinated HBDI⁻ structures as reference geometries. These are shown as red circles with a plus together with the MECI-I (yellow diamond with a plus). (d) Normalized distributions of velocity components for the parent TBF along the *h*-direction (dominated by I-torsional motion, see Fig. S9) for the earliest non-adiabatic transition for each initial condition events along the I-twisted intersection seam. (e) Normalized distributions of S₁/S₀ energy gaps for the non-adiabatic transition events along the I-twisted intersection seam. Events for both positive and negative directions have been combined.

P17, para 2, topic (ii). It is not clear that these three sentences are all concerning the same substituent effect under (ii) - can the connection be made clearer or section rewritten?

Thank you very much for pointing this out. We have revised the paragraph to better connect the three points to the effects of trifluorination and remove unnecessary details.

Revised version:

Based on the desired I-twist favoring changes caused by 2,3,5-trifluorination, we further simulated its dynamical impact upon photoexcitation. ~~Our results can be summarized in three key substituent effects~~ The effects of trifluorination can be summarized in three key points: (i) it essentially turns off the unproductive P-twisting channel and hence increases the propensity of the system to progress along the potentially reactive pathway. This is a result of an increase in the effective torsional excited-state barrier through both potential and inertial effects; (ii) it accelerates internal conversion via the I-twisted pathway by changing the energetic location of the minimum on the intersection seam to be in proximity of the twisted S₁ minimum. ~~It leads to a stabilization of the |P> diabatic state, characterized by charge location on the P-ring.~~ This changes the direction and velocity of approach to the I-twisted CI seam, which is no longer gated by the HOOP motion; (iii) ~~as such, trifluorination induces~~ a more ballistic behavior along the isomerization-driving coordinate upon approach to the I-twisted CI seam nearly doubles the photoisomerization quantum yield (from ~30 to ~60%).

The discussion mentions specific H-bond structures which may have an impact when the chromophore is in the protein. It would be helpful to mention and speculate on the effects of nonspecific 'solvation dynamics' in polar/nonpolar fluid solvents? These are typically driven by the polar solvent response to permanent dipole moment (and others charge distribution) changes on excitation. How might these effects modify relaxation through one or other pathway. Such polar solvent dynamics seem to have only a modest effect on HBDI photophysics.

We have in previous computational work (Jones, List, Martínez, Chem. Sci. 2021, 12, 11347) investigated the excited-state behavior of HBDI⁻ in aqueous solution. In an equilibrium picture, the solvent effects are indeed nonspecific, leading to similar solvent-stabilization of both the I- and P-twisted (charge-separated) S₁ minima. However, our nonequilibrium dynamics simulations indicate that the equilibrium solvation structure for the I-twisted S₁-minimum resembles that of the ground state solvation much more closely compared to the P-twisted S₁-minimum solvent structure (see Figure 11 in our previous work and fragment charges in Table S14 of the present work). Accordingly, the I-twist biased solvent arrangement in the ground state seems to explain the observed nonequilibrium response that prefers the I-twist pathway (70% of the population) over the P-twist pathway (30%) for the aqueous HBDI⁻.

In this work, we investigate the possibility for selective stabilization of one of the torsional pathways by chemical substitution with the aim of enabling control of the excited-state dynamics. Considering TFHBDI⁻, we expect that the non-selective charge-transfer-stabilizing water could potentially open the P-twist channel to some degree, counteracting the desired I-twist selectivity. On the other hand, the stabilization of charge on the P-ring in TFHBDI⁻ due to the electron-withdrawing capacity of the fluorine atoms (see Table S14) suggests that the ground-state solvent structure for TFHBDI⁻ will be comparatively more biased toward the I-twist pathways in the nonequilibrium dynamics. In our view, these are interesting speculations that can be addressed computationally but merit a future study focused on combined substituent and solvation effects. For this reason and to keep the focus on pathway selectivity, we have decided against including such discussions in the present work.

Finally, it would be very interesting to extend these studies to the neutral chromophore, for which a moderately efficient E-to-Z isomerization is the key step in reversibly photoswitchable proteins, and which generally has much richer photochemistry (including photoacid response) than the anion.

Page 10 of 13

We thank the reviewer for the enthusiasm around the neutral chromophore. We agree that it would be interesting to target this system in future studies both in solvated and protein settings.

Reviewer 3:

This work investigates the isomerisation of GFP chromophores using multireference methods. Three different chromophores are studied in their anionic form in vacuo. The canonical chromophore was compared to two chemically modified HBDis that were experimentally introduced by Boxer and coworkers. The main question is which of the two potential double bonds is selected for isomerisation and how the chemical modification will influence the selection.

The work is extensive (conical intersection optimization, minimum energy path and nonadiabatic dynamics) and done at the highest possible level. However, I found some portions of the text a bit difficult to follow and had to re-read them several times.

We thank the reviewer for pointing out these aspects that were not sufficiently clear. We have rephrased and clarified various aspects to improve the clarity of the manuscript.

The manuscript should be accepted after some minor revision which is mainly for clarification:

1) A reduced active space was established for the canonical GFP chromophore. Can the authors explain why it is still sufficient for the modified chromophores especially for the fluorinated version which is involved in pi-electron donation? The methoxy substituted version can also extend the pi-conjugation of the aromatic ring via the oxygen.

A full π -active space would consist of the p_z -orbitals for each heavy atom (17 for TFHBDI⁻, excluding the methyl C-atoms). However, a full π -active space is not necessary (nor desired in dynamics simulations for active-space stability reasons) to capture the electronic character of the S_1 state, which for TFHBDI⁻ and MHBDis remain dominated by an excitation involving the methine bridge π -orbitals (bonding, non-bonding and antibonding). In a reduced active space calculation, the effect of the substituent enters mainly at the mean-field level. In other words, leaving out the fluorine p-orbitals of the active space (as for any of the occupied and secondary orbitals) will reduce the amount of dynamical electron correlation that is captured. In our calculations, we partially correct for this either in a second-order perturbative fashion (XMS-CASPT2) or using the empirical α -correction (α -CASSCF). For substituents with π -orbitals energetically close to the bridge π -orbitals (e.g., nitro groups), the active space will likely have to be expanded to capture the electronic states because of “interfering states” directly involving the π -orbitals on the substituents.

2) The minimum energy pathways were computed for the in-plane and out-of-plane

orientation of the methoxy groups separately. Is it possible for the orientation to change during the photoreaction? Was this observed during NAMD?

The most stable ground-state conformation of the methoxy group is the out-of-plane orientation. In the test simulations we ran for MHB DI^- , we observed limited oscillations of the methoxy group (between the out-of-plane and the in-plane forward-pointing transition state orientation, see Figure S4) for the initial conditions proceeding along the I-twist pathway. For the initial conditions following the P-twist pathway, we observed a larger tendency for the methoxy group to undergo full rotation, such that it also visits the in-plane orientation with the methoxy group pointing away from the phenol O-atom. In this work, we however focus on TFHB DI^- dynamics because of its desired I-twist promoting effect, and hence, we did not consider these aspects further.

3) Recent studies on related chromophores have suggested the possibility of a combined isomerization of the single and double bond. Was this observed in NAMD simulations?

In the gas phase, the two bridge-torsional modes are largely decoupled (see Figure S6 in our previous work, List, Jones, Martínez, Chem. Sci. 2022, 13, 373). The same holds for the substituted derivatives. In other words, we essentially only observe one-bond-flip dominated motion. The torsional motion becomes somewhat more coupled in aqueous solution (Figure 5, Jones, List, Martínez, Chem. Sci. 2021, 12, 11347). Increased torsional coupling may also be expected in a protein setting due to the confining environment and volume-greedy one-bond-torsions (particularly along the I-twist mode), but the specific deactivation mechanism will be highly system-dependent (for GFP and Dronpa2 behaviors, see e.g. Figures S17/S18, Jones, List, Martínez, JACS, 2022, 144, 12732).

4) There is an inconsistency in Fi. 4. It looks like the S1-planar geometry of TFHB DI^- is lower in energy than the S1-P minimum, but in fig. 3 its higher. Is the level of theory the same?

The NEB paths in Figure 4 were generated at the α -CASSCF(4,3) level while the energies were computed at the XMS-CASPT2 level. This means that the geometries used in Figure 4 are not the exact same as the XMS-CASPT2 critical points in Figure 3 (all XMS-CASPT2). In the latter case the difference between S1-P and S1-planar is -0.03 eV whereas in the former case (α -CASSCF NEB structures), it amounts to +0.04 eV. In both cases, the driving force toward P-torsion characteristic of HB DI^- is removed. To highlight this difference in level of theory (between energies and structures), the original text stated: "Fig. 4 shows estimates for XMS-CASPT2 S₁ barriers (in mass-weighted coordinates) between the planar S₁ geometry and the twisted minima as computed based on nudged-

elastic-band (NEB⁴⁴) paths obtained at the $\alpha(0.64)$ -CASSCF(4,3)/6-31G* level of theory. As such, these values represent upper bounds to true barriers.“

To avoid confusion, we have now clarified these differences in the caption of the new figure 3 (former figure 4).

New figure 3:

Fig. 3. Relative XMS-CASPT2(4,3)/6-31G* energies along the S_1 minimum energy path connecting the S_1 -planar geometry to each of the two twisted minima (left: S_1 -P, right: S_1 -I). The path was computed using NEB at the α -CASSCF(4,3)/6-31G* level and hence the end points do not exactly match those in Fig. 2. While the barrier estimates for I- and P-twisting are approximately the same in HBDI⁻ (gray), leading to almost equal bifurcation along the two competing pathways, both potential and inertial effects of 2,3,5-trifluorination (orange) act in concert to remove the driving force along the P-torsion in TFHBDI⁻. The effect of 3-methoxylation (blue) depends on the orientation of the methoxy substituent. In the ground-state favored out-of-plane orientation (solid line), the relative energetics of the twisted S_1 minima and barrier estimates resemble those of HBDI⁻. In the in-plane orientation, the two twisted S_1 minima become nearly isoenergetic with a small barrier of (~ 0.05 eV) along the I-torsion.

REVIEWERS' COMMENTS:

Reviewer #1 (Remarks to the Author):

I am satisfied with the changes made by the authors to the revised version of the manuscript. I now fully support its publication in Communications Chemistry.

Reviewer #2 (Remarks to the Author):

All of the points raised have been properly addressed. I look forward to seeing the work in print.

Reviewer #3 (Remarks to the Author):

The authors have addressed all concerns and comments. I recommend the manuscript for publication.